# Estimating the Workability of Concrete with a Stereovision Camera during Mixing

**DOI:** 10.3390/s24144472

**Published:** 2024-07-10

**Authors:** Teemu Ojala, Jouni Punkki

**Affiliations:** Department of Civil Engineering, School of Engineering, Aalto University, 02150 Espoo, Finland; jouni.punkki@aalto.fi

**Keywords:** concrete, workability, computer vision, stereovision, machine learning

## Abstract

The correct workability of concrete is an essential parameter for its placement and compaction. However, an absence of automatic and transparent measurement methods to estimate the workability of concrete hinders the adaptation from laborious traditional methods such as the slump test. In this paper, we developed a machine-learning framework for estimating the slump class of concrete in the mixer using a stereovision camera. Depth data from five different slump classes was transformed into Haralick texture features to train several machine-learning classifiers. The best-performing classifier achieved a multiclass classification accuracy of 0.8179 with the XGBoost algorithm. Furthermore, we found through statistical analysis that while the denoising of depth data has little effect on the accuracy, the feature extraction of mixer blades and the choice of region of interest significantly increase the accuracy and the efficiency of the classifiers. The proposed framework shows robust results, indicating that stereovision is a competitive solution to estimate the workability of concrete during concrete production.

## 1. Introduction

Concrete has various advantages over other construction materials, such as high compressive strength, excellent durability, high fire resistance, and a high level of local manufacturing. After mixing in a concrete plant, ready-mix concrete is transported to the construction site and compacted into various shapes and types of concrete structures. One critical property of fresh concrete is workability, which describes how easy the concrete is to place and compact [1]. Concretes that do not meet the workability requirements can be rejected during the on-site inspection, causing extra waste and expenses [2] and delaying the casting process. It has also been demonstrated that the risk of segregation significantly increases during compaction when more fluid concretes are used [3]. Consequently, variation in concrete workability can have harmful effects on the quality of the concrete structure. Therefore, measurement systems must be developed for the fresh concrete, ensuring that the concrete conforms to the set requirements.

Concrete comprises constituents whose natural variation causes changes in fresh concrete properties. Since a significant part of the concrete consists of low-processed natural aggregates, grading and moisture content of aggregate vary and cause variations in the workability during the batching process [4,5,6,7]. The grading and moisture content of the aggregate is measured in the mixing station to adjust the mix composition to compensate for this natural variation. However, significant fluctuation in workability may remain even with successive concrete batches proportioned with the same recipe. Moreover, increasingly more complex mix compositions are required to satisfy the future requirements of high-performing green concretes [8,9]. These more demanding mix compositions can create challenges in reaching the desired properties for fresh concrete. Continuous measurement systems could detect defective concrete in real time, decreasing the safety margins in concrete production and reducing costs and emissions.

Rheological properties have been used to describe the workability of concrete [1,10] and to estimate concrete behavior during its placement into the formwork [11]. The fluid behavior of concrete is often described using the Bingham model [12], where parameters plastic viscosity μ (in Pa∙s) and yield stress τ_0_ (in Pa) depict the flow characteristics. As such, Bingham fluids should be characterized using two-point tests [1], where the relationship between the viscosity and yield stress are determined at least in two points. Because rheological testing requires special equipment, it has been used only as a measurement method in concrete research. Consequently, the correct workability is verified using a one-point test called the slump test [13], which describes how concrete behaves under gravitational force [1]. Thus, the slump test does not sufficiently describe the behavior of concrete, as concretes with the same slump value can have different rheological properties. While the testing equipment is lightweight and technically rudimental, only a fraction of the concrete batches are tested due to the high cost of labor and time. Moreover, the test is susceptible to incorrect execution and reporting. Therefore, new approaches are needed to improve concrete production quality control and safety assurance [9,14]. These techniques should overcome the limitations of one-point tests and allow continuous and real time measurement possibilities.

Various machine learning (ML) methods have been used to characterize concrete, optimize mix designs, estimate fresh and hardened concrete properties, and investigate concrete durability and cracks, as summarized exhaustively in [15,16]. In addition, the application of ensemble ML models has improved the accuracy and robustness of these models, such as eXtreme Gradient Boosting (XGBoost) which leverages tree boosting [17,18]. Consequently, the scalability and superior performance of these models have proved their capabilities in the field of engineering [17,18,19,20,21,22]. The high performance of XGBoost can be explained by its efficiency in tree pruning, regularization, and parallel task processing [18]. To predict the workability of concrete, the ML models have relied on numerical data, such as tabular information about concrete composition [23,24,25,26,27]. However, predicting the workability of concrete based on the concrete composition is always vulnerable to errors caused by the natural variability in constituents. Consequently, the variability should be minimized during the mixing process, which cannot be achieved by monitoring the numerical data alone.

A few computer-vision (CV) techniques have emerged to estimate concrete workability based on digital images. Li and An [28] first used CV to predict the workability of self-compacting concrete (SCC) using images captured from a single-shaft laboratory mixer. By analyzing these images, they quantified the shape of the concrete on the mixing blade, which correlated with the slump-flow test results. While converting the visual information into features demonstrated promising results, the extraction procedure required significant effort. Similarly, Ding and An [29] used the same visual information to estimate the moisture content of SCC concrete during the mixing process. This method relied on the linear correlation between the slump-flow test and the water-to-powder ratio when the amount of superplasticizer is constant. The study demonstrated how CV techniques can be used to optimize the mix compositions if the other parameters are known.

Deep learning (DL) models can be used to predict the workability of SCC, as discovered by Ding and An [30,31]. In addition, sequential image information can be fed to the convolutional neural network (CNN) model when long-short-term memory (LSTM) [32] units pass the temporal information forward in the network. The framework involved recording videos during the mixing process and pre-processing them into short sequences before the model training. Alternatively, Yang et al. [25] used CNNs to identify multiple characteristics of the mix composition from digital images. The images depicted different concrete batches placed on a large tray after mixing. Unfortunately, the DL models are challenging to interpret, making identifying the essential features for good estimation difficult.

Coenen et al. [33,34] developed a new framework that enables the estimation of concrete workability from the concrete truck during the discharge process. In their research, the flow characteristics were predicted from observing sequences of images of flowing concrete in open-channel geometry, representing a chute of a concrete truck. The images were converted into spatio-temporal flow fields that were used as input for the CNN model. The model estimated both the consistency and rheological parameters of the concrete. As noted by the authors, the flow field of the concrete mixer is more erratic, making the framework less likely to function in the concrete mixer. Moreover, predicting the workability of stiff concretes may be more difficult since the surface of the concrete flows more uniformly. In these DL approaches to estimate the workability in the mixer, the models see only a two-dimensional presentation of the concrete.

A key component of human visual estimation of workability is the perception of depth. Tuan et al. [35] proposed a stereovision system to identify the slump value after lifting the cone in the standardized slump test. The computed depth maps provided the slump of the concrete. In addition, they used the DL model to classify concrete slumps so that only valid test results were measured. Yoon et al. [36] combined point cloud analysis extracted from the standardized mini-slump flow test using a depth camera and an artificial neural network (ANN) to estimate fresh concrete properties directly from the concrete sample’s diameter, height, and curvature after the slump cone has been lifted. While this approach does allow automatic logging of the results, removing the user error when reading the slump value, it does not remove the tedious setup of the testing process. Ponick et al. [37] have also used stereovision to determine the rheology of ultrasonic gel through a mixing paddle. Their work established a convincing correlation between rheological parameters and stereo images using a flow curve. This curve was obtained through the training of the CNN model based on the 3D reconstruction of the substrate’s surface. The study demonstrated the potential of depth measurement and the possible relationship between the substrate and rheological parameters, which can be used to relate depth information to rheological parameters if required.

In this paper, we propose a novel framework to estimate the concrete slump using a stereovision camera during the mixing process. Unlike previous studies on workability estimation, this research utilizes depth data collected during the mixing process to train several ML classifiers while maintaining a high level of transparency across the whole pipeline. Furthermore, the depth data are feature-engineered to improve the efficiency of the framework, focusing on the intrinsic features of the concrete mixer. The proposed framework consists of data collection, data preparation pipeline, training, and prediction of the ML models, and finally, model selection based on the performance. The main contributions of this study are:We designed an ML framework to estimate the slump of concrete using a stereovision camera during the mixing process.We established a transparent and lightweight ML pipeline to convert the depth information to suit the ML algorithms while preserving the core features for estimation of workability.We investigated the robustness and sensitivity of the framework through comparative and statistical analysis between different data preparation methods.

## 2. Materials and Methods

### 2.1. Materials and Mix Composition

A concrete mixture composition was batched in laboratory conditions where the temperature of the materials and the laboratory was 20 °C ± 2 °C. The mix composition of the concrete was designed so that only a part of the water of the mix composition could be added during the initial mixing to produce stiff concrete with a slump class of S1. The remaining water was added in multiple steps during the experiment without introducing segregation effects to the concrete. It was experimented that adding 0.341 kg of water in each mixing cycle increased the slump class from S1 to S5, increasing the water-to-cement ratio from 0.40 to 0.55. Thus, a mix composition with the final water-to-cement ratio of 0.55 was designed, consisting of 390 kg/m^3^ of cement, 1698 kg/m^3^ of aggregates, 215 kg/m^3^ of effective water, and 3.9 kg/m^3^ of superplasticizer (MastersBuilders, MasterGlenium Sky 600, Riihimäki, Finland). A common cement type CEM II/B-M S-LL 42.5 N was used in the mixture, produced by Finnsementti (Parainen, Finland). Seven aggregate fractions were used, which were washed and sieved and had a maximum aggregate size of 16 mm.

### 2.2. Experimental Setup

A planetary laboratory mixer (MP 75/50, Sicoma, Perugia, Italy) was used to mix the concrete batch, as shown in Figure 1a. The volume of the batch was 35 L, which is the maximum volume recommended by the manufacturer. The forced mixing system of the mixer consists of two blades; an arm-bearing mixing spider blade that rotates around its axis as well as the axis of the mixing tank and a second arm blade on the opposite side of the spider, which scoops the concrete towards the center of the mixing tank. The rotation speed of the main shaft was set to 44 rounds per minute.

Six mixing cycles were performed to collect the required slump classes. The dry constituents were mixed for 30 s during the first mixing cycle. The water was poured into the mixer within 30 s without stopping the mixer. Similarly, a superplasticizer dosage was added to achieve the initial slump target within the next 30 s and finally mixed for another two minutes. The slump test was performed immediately after each mixing cycle. A new mixing cycle was started every 15 min by adding the additional water dosage into the mixer. The mixture was mixed in the following cycles for two minutes to allow the concrete to become homogenous first and then to collect a sufficient amount of depth data. Because the slump class did not increase during the fifth mixing cycle, an additional mixing cycle was added to achieve the slump class of S5.

An active stereovision camera (RealSense™ Depth Camera D455, manufactured by Intel, Santa Clara, CA, USA) was used to collect the depth data, as shown in Figure 1a. The stereovision camera was placed on a standalone support, isolated from the vibrations of the mixer. This camera uses an additional infrared light source between the two cameras to assist the camera in computing the depth without a visible light source, making the camera less sensitive to external lights and the angle of incidence. Furthermore, the camera has a global shutter to minimize the effect of a rolling shutter when recording high-speed motion, such as rotating mixer blades. The stereovision camera uses parallax and triangulation to estimate the depth [38]. A small distance between the cameras allows the triangulation of the object’s position and translates that into distances between the camera and the subject. A stereovision system typically has two parallel cameras with the same focal length (*f*). The distance between the cameras is called the baseline (*b*). The difference in the position between left (*x_l_*) and right (*x_r_*) frames for an object is called disparity (*d*). Hence, the distance to the object (*Z*) from the camera is calculated using Equation (1):(1)$$Z=\frac{bf}{d}=\frac{bf}{x_{l}-x_{r}}.$$

In our experiment setup, the stereovision camera was placed on an arm connected to a separate column to isolate it from the vibrations of the mixer, as seen in Figure 1a. With a fixed field of view of 87° × 58°, the camera could cover approximately one-third of the mixer floor area, as seen in Figure 1b. A pre-calibrated accuracy of the depth camera was reported to be <2% at four meters. During the experiment, the camera was tethered to a laptop where Intel RealSense™ Viewer software (2.43.0) was used to select the camera settings and to record the depth data via a high-speed USB cable. The preview of the depth data can be seen in Figure 1c, where objects closer to the camera are rendered with white pixels, and objects further away are rendered with darker pixels.

### 2.3. Data Preparation

The depth data was prepared for machine learning using the developed processing pipeline, as summarized in Figure 2. The pipeline was divided into three main phases: (1) denoising of the depth data, (2) feature extraction, and (3) preparation of datasets. The depth data was stored as one unsigned 16-bit integer per pixel (Z16) format and cannot be viewed without conversion to a standard video format such as MP4 file format. Many filtering methods have been developed to denoise the depth data from artifacts, triangulation errors, and temporal noise. The authors experimented with various denoising methods integrated into the Intel RealSense ROS wrapper. These methods are also available in the Intel RealSense Viewer software. The authors followed the recommended scheme for the processing pipeline and its RealSense tools, which the manufacturer presented. The code was implemented in Python 3.7 and is available upon request. The data preparation phases are described in more detail in Section 2.3.1, Section 2.3.2 and Section 2.3.3.

#### 2.3.1. Phase 1: Denoising of the Depth Data

Two denoising methods (DM 1 and DM 2) were applied to process the depth data in the first phase. While DM 1 represents minimal denoising (Figure 3a), DM 2 represents substantial denoising of the depth data (Figure 3b). Figure 3c shows the denoising process where depth frames are denoised using various filters, resulting in filtered depth frames. It can be seen in the figure that DM 1 skips the denoising filters, only applying a custom color map. On the other hand, the denoising techniques in DM 2 consisted of decimation, threshold, spatial, and temporal filters. The most prominent denoising filter in DM 2 is a hole-filling filter, which fills the areas where the triangulation failed, shown as black pixels in Figure 3a. This morphological filtering has been used to remove noise caused by the camouflage and lighting effects [39]. The filtered depth data were saved as grayscale videos, consisting of frames where the depth measurements are seen as pixel intensity values in a range between (0 and 255), depending on the distance to the camera as set by the threshold filter. By selecting a low limit of 45 cm and a high limit of 90 cm, the mixer floor was seen in near black and the blades in near white pixels in the resulting videos.

#### 2.3.2. Phase 2: Feature Extraction

In the second phase, feature extraction techniques were applied to the grayscale videos. The feature extraction aimed to potentially improve the performance of the models as well as investigate the robustness and sensitivity of the ML classifiers. As shown in Figure 2, the extraction phase was divided into three steps, where the videos were processed to find ideal features for the ML models. The extraction started with a selection of three different regions of interests (ROIs), which were 150 × 150 px^2^, 100 × 100 px^2^, and 50 × 50 px^2^. The resulting images are visualized in Figure 4, where an example frame was cropped based on the three ROIs. The figure illustrates how the largest ROI maximized the visible surface area, while the smaller ROIs focused on the small area where the mixer blades interacted with the concrete.

In the next step, four filtering methods (FMs) were applied to highlight the features of the mixer: FM 0 was a reference method without denoising, FM 1 removed the frames with mixer blades visible, FM 2 kept the frames with the scoop visible, and FM 3 kept the frames with the spider blade visible. Before the denoising, the frames were divided into rotation cycles based on the visibility of the yellow mixer blades by monitoring the number of yellow pixels in each frame.

Haralick texture features (HTFs) have been used to convert visual data into numerical values in increasing numbers in various fields, as summarized by Löfstedt et al. [40]. These features were first used by Haralick et al. [41], who proposed using a gray-level co-occurrence matrix (GLCM) to compute statistical features. HTFs represent statistical properties such as contrast and entropy [41] and are known for their simplicity and intuitive interpretation [40]. They are also believed to be robust against noisy images [42]. This study used the Mahotas library for Python (Mahotas 1.4.13) to compute HTFs for each frame. The GLCM and Haralick texture features can be computed using definitions in Table 1, as described in [40,41]. In the table, the P(i,j) describes the elements i,j in the unnormalized GLCM, and the *N_g_* represents the number of grey levels in the quantized image [40,41].

#### 2.3.3. Phase 3: Preparation of Datasets

The datasets were generated for each DM, ROI, and FM combination. Thus, 24 unique datasets were created for the training and evaluation of the ML models, as shown in Table 2. Only the last 40 s of each mixing cycle was used to ensure a sufficient representation of data, which resulted in an unfiltered number of 2400 frames for each slump class. Consequently, the reference filtering method (FM 0) dataset consisted of 12,000 frames, or observation points after the conversion to HTFs, for all five slump classes. For the remaining FMs, the number of observation points varied between 2854 and 8117 in the datasets, depending on the amount of filtration. The observation points were divided into training and testing datasets by selecting the last 20% of the frames for the testing. While the training dataset was used to train and validate the ML classifiers in the training process, the testing dataset was only used in model selection. Since the ML classifiers required the output labels in numerical format, the slump classes were numerically hot encoded as follows: S1: 0, S2: 1, S3: 2, S4: 3, S5: 4.

### 2.4. Training of Machine Learning Classifiers

#### 2.4.1. Selection of ML Classifiers

The investigated ML classifiers are summarized in Table 3, but more detailed information about these classifiers can be found in [43]. The table shows the abbreviations and working principles of the nine selected ML classifier algorithms: decision tree (DT), gradient boosting (GBoost), k-nearest neighbor (KNN), logistic regression (LR), multi-layer perceptron (MLP), naïve bayes (NB), random forest (RF), and support vector machine (SVM) and extreme gradient boosting (XGBoost), from which GBoost and XGBoost algorithms were so-called ensemble algorithms. The Python library Scikit-learn (0.24.2) was used to initiate these algorithms in the training process, except for the XGBoost model, the employed Python library XGBoost (2.0.3). The performance of the ML model is highly influenced by a combination of hyperparameters [19,44]. As such, applicable hyperparameters were investigated extensively to find the best-performing configuration for each classifier. As a result, the best 216 models were selected from a total of 9778 trained models, each selected model representing the unique combination of dataset and ML classifier.

#### 2.4.2. Evaluation Approach

Various evaluation approaches were employed to assess the framework presented in this paper. For the training process of the ML classifiers, the datasets were divided into training and testing datasets. The training dataset was used with cross-validation (CV) to train and validate the models. The CV allows a more generalized evaluation of the model’s performance when the dataset is limited [57]. As such, CV has been used a lot in concrete sciences [58,59]. The most used technique, k-fold CV, runs multiple iterations for model validation, resulting in a more robust estimation [15]. For the paper, a k-fold inner CV was selected (see Section 2.3.3) with five splits (k = 5). It must be noted that k-fold CV can still introduce a bias with smaller datasets, which is often the case with concrete studies [57].

After the training, the model performance was investigated by computing the evaluation metrics that are commonly used to assess the performance of the ML classifiers. They provide quantitative insight for hyperparameter tuning and model selection. The Python library Scikit-learn (0.24.2) was used to compute the PCA and performance metrics. Before this, each instance is classified as true positive (*TP*), true negative (*TN*), false positive (*FP*), and finally false negative (*FN*), as listed in [60]. The principles of calculating accuracy, precision, recall, and *F*1 score are shown in Equations (22)–(25). The authors suggest a paper by Hossin and Sulaiman [61] for a more complete review of evaluation metrics for classification purposes. In addition, a quadratic weight kappa (*QWK*) [62] was computed to analyze the agreement between the instances using Equation (26). The accuracy is calculated using the equation:(22)$$Accuracy=\frac{TP+TN}{TP+FP+TN+FN}$$
where the accuracy represents the ratio of correctly predicted instances to all evaluated instances. In the equation:(23)$$Precision=\frac{TP}{TP+FP},$$
where the precision represents the ratio of correctly predicted positive instances to all predicted positive instances. The recall presents the ratio of correctly predicted positive instances to all actual positive instances in the equation:(24)$$Recall=\frac{TP}{TP+FN}.$$

The precision and recall metrics can be described with the *F*1 score that represents the harmonic mean between the precision and recall with the equation:(25)$$F1\;score=2\cdot \frac{Precision\;\cdot Recall}{Precision+Recall}$$

To analyze the agreement between a set of predictions and a set of multi-class labels, *QWK* was computed using the equation:(26)$$QWK=\frac{p_{0}-p_{e}}{1-p_{e}}$$
where *p*_0_ represents the observed agreement ratio, *p_e_* represents the expected agreement of randomly assigned labels.

In addition to the evaluation metrics, the performance was visualized with computing confusion matrices, which reveal the disposition of the predicted and actual instances [60]. In other words, the confusion matrix provides information on how all the predicted instances are compared to the ground truth values, e.g., measured slump classes. As noted by Fawcett [60], the values along the diagonal line of the confusion matrix show the correct predictions made by the classifier, and the values of this diagonal reveal the mispredictions between the various classes.

A principal components analysis (PCA) statistical technique was performed on the dataset with the best-performing dataset (DM 1, ROI BIG, FM 0) to uncover the patterns in high-dimensional data. The PCA technique aims to reduce the dimensions of the dataset while maintaining as much variability as possible, making the structure of high-dimensional data more interpretable with minimal information loss [63]. This reduction is achieved by finding new uncorrelated variables, e.g., principal components (PCs), that consecutively maximize variance for the given dataset. In addition, a biplot was drawn to display both the transformed values of the HTFs and the loadings for the PCs. The biplot helps to explain how HTFs contribute to the variation captured by the PCs and how they are related to these components. In the biplot, the HTF vectors point in the direction of the maximum variance, revealing how a particular feature influences the PCs. The arrow length reflects the importance of a feature in explaining the variance captured by the PCA. As such, a longer arrow means that the feature plays a more significant role in the data variance along the direction of a principal component. Vectors pointing in the same direction indicate a strong correlation between the features, and vectors aligning with the PC axis are highly influential to that component.

Finally, the impact of the data preparation was analyzed by assessing the performance of the ML classifiers with different parameters through visual comparison and statistical analysis. The three data preparation parameters were selected for this analysis: DM, ROI, and FM. The testing results were grouped based on the nine selected ML classifiers. A Shapiro–Wilk test [64] was first conducted to confirm the normal distribution of the data. Since the normality assumption was violated (*p*-value < 0.05 with α = 0.05), the null hypothesis was rejected, and non-parametric tests were executed, which do not require the assumption of normal distribution across the independent groups. While the Mann–Whitney U test [65] was performed to compare DM, the Kruskal–Wallis test [66] was carried out to compare groups within the ROI and FM. The tests produce a statistic from which a *p*-value can be derived. A low *p*-value indicates a statistically significant difference between the medians of the groups. Additionally, Dunn’s Multiple Comparison test [67] was carried out to indicate the specific groups causing the difference if a statistical difference was found between three or more groups. The tests were computed using a Python library, SciPy (1.7.1).

## 3. Results and Analysis

### 3.1. Fresh Concrete Results

The slump test was performed after each mixing cycle. Table 4 shows the measured slumps and their corresponding slump classes. In addition, the table notes the range of slump values allowed within each slump class as presented in SFS-EN 206 [68]. The fifth mixing cycle was repeated, as the water addition did not increase the slump class to S5.

### 3.2. Model Selection by Performance

#### 3.2.1. Performance Metrics

The evaluation results of the nine ML classifier algorithms are shown in Table 5. The most accurate model across all the metrics was trained with the XGBoost algorithm, reaching a testing accuracy of 0.8179. The table shows that the model was trained with dataset parameters of DM 1, ROI BIG, and FM 2. Furthermore, models trained with algorithms MLP (0.8078), GBoost (0.7991), and RF (0.7855) achieved reasonable testing accuracies. The lack of differences between the calculated evaluation metrics such as testing precision, recall, and thus F1 score, suggests that models have consistent model performance, such as predicting false positives and negatives, implying that models do not favor certain classes. A higher QWK was consistent across all classifiers, revealing that the multi-classification problem contains ordinal classes where the models tend to confuse neighboring classes from the true class. By penalizing these misclassifications less, the QWK score of 0.9156 was reached with XGBoost. In addition, a similar performance indicates that models are robust in generalizing the training data to the unseen test data.

#### 3.2.2. Confusion Matrices

The confusion matrices for each best classifier are visualized in Figure 5. The confusion matrices reveal that the predictions are mainly located along the diagonal line with all classifiers, and only a fraction of the predictions were mislabeled with algorithms such as XGBoost, MLP, and GBoost. It can be observed that the models are misclassified more with stiff concrete, indicating that the models perform better with medium to high workability concretes. This phenomenon is more pronounced with worse-performing classifiers such as LR and SVM in the figure. The authors believe this could be linked to a lack of deformation with stiff concrete caused by the higher yield strength. When the yield strength is sufficiently low, the concrete starts to move without restrictions governed by its viscosity.

Figure 5 further reveals that the incorrect predictions seem to be located in the neighboring slump classes, indicating that the classifiers have more difficulties distinguishing similar workabilities. This phenomenon is probably caused by discretizing the slump values into slump classes. There is also a possibility that the shape of the concrete surface resembles frames collected with different slumps at specific moments during the mixing cycle, causing greater misclassifications. However, the number of incorrect predictions outside the neighboring classes is negligible with the best-performing models. A regression-based prediction model would better reflect the nature of consistency without having hard limits between the slump classes.

### 3.3. Principal Component Analysis of the Haralick Texture Features

The PCA technique was performed to inspect how the HTFs relate to the slump classes. The dataset (DM 1, ROI BIG, and FM 2) for the best-performing model was selected for the analysis, consisting of 6939 computed Haralick features as observation points. The number of PCs was selected based on Kaiser’s rule, stating that the principal components (PCs) should have greater than or equal power to explain the variance in the data [69]. Thus, two PCs were retained with an eigenvalue less or equal to one, according to the scree plot in Figure 6a.

The values of the two main PCs are plotted in Figure 6b, where they are color-coded according to the measured slump classes, each representing an individual frame. From the figure, different slump classes can be distinguished as overlapping clusters concentrated in different locations. In more detail, fluid concrete tends to have lower PC 1 values, whereas stiffer concrete occupies higher values of PC 1. The values of the PC 2 seem to increase as the concrete becomes more fluid. Interestingly, the variability in PC 2 appears to decrease with the fluid concrete, which might be correlated to a less complex but more repeatable concrete appearance.

The relationship between the HTF vectors and PCs can further reveal how well the PCs can describe the collected depth data. In Figure 6b, the vectors representing HTFs from f_1_ to f_13_ (presented in Section 2.3.2) indicate that all HTFs contribute to the variance in the data since all the vectors, except vector f_3_, are similar in magnitude. Furthermore, the direction of the vectors seems to follow the axis of the PCs, except with the vectors f_9_ and f_13_, suggesting a high correlation between PCs and the HTFs. Haralick et al. [34] speculated that some visual properties can be expected from certain textural features. For example, the level of entropy in an image may increase as the complexity of the image increases. The loss of entropy (f_8_, f_9_, and f_11_) with fluid concretes can be explained by the smoother appearance caused by their tendency to self-level under the gravitational force. In contrast, stiff concrete with higher yield strength causes more exaggerated depth differences that increase the perceived entropy. The values with stiff concrete were more scattered along the PC 2 and the HTF vectors representing the contrast (f_2_) and the variances (f_4_ and f_7_). A wide separation between dark and light pixels is intrinsic to high-contrast images, leading to a high variation in the grey-level values. Hence, these vectors indicate that contrast decreases when the concrete becomes more fluid. This minor variation with fluid concretes can be explained by the low plastic viscosity, which allows the surface to deform during the mixing process.

### 3.4. Impact of Data Preparation

To investigate the impact of the data preparation on the performance of the classifiers, comparative and statistical analysis was performed on the trained models. The dataset preparation parameters (DM, ROI, and FM) were investigated separately in three steps. For each preparation parameter, all the models were divided into groups determined by the number of methods in each preparation parameter. 

Figure 7 compares how each classifier performs under the different DMs. Based on the figure, there are no apparent differences between DMs. Correspondingly, the Mann–Whitney U test confirmed an insignificant difference (*p* = 0.5750) between the medians of the groups (Shapiro–Wilk test, *p* < 0.05). However, a slightly higher variation in the testing accuracies with DM 1 suggests that applying DM 2 might improve the quality of the data and assist in generalizing the models by reducing the number of errors in the depth images. Therefore, the DM might positively affect the performance of the models for specific scenarios where the number of artifacts is substantial.

Figure 8 displays the testing accuracies across the FMs for each classifier. Based on the results, the FMs significantly affected the average performance of the classifiers (Kruskal–Wallis test, *p* = 0.0046). The reference method (FM 0) demonstrated the poorest average performance within most classifiers, indicating that feature extraction, by removing certain frames, improves the ML model performance. When the models were presented with frames with the scoop blade (FM 2), the testing accuracy increased with all the better-performing models, especially when compared to the reference method (Dunn’s test, *p* = 0.0049). Interestingly, focusing on the spider blade (FM 3) instead of the scoop blade negatively affected the performance of these classifiers (Dunn’s test, *p* = 0.0290). The authors believe that the movement caused by the spider blade might be too complex or erratic to collect core features for prediction, limited by the accuracy and resolution of the depth data. Consequently, the movement of the concrete when the scoop blade sweeps the concrete is more suitable for predicting slump using the stereovision camera. 

Figure 9 shows the testing accuracies for each classifier with the three ROI sizes. Based on the figure, the classifiers performed best with ROI BIG, which limited the view on the edges of the mixer. Kruskal–Wallis test confirmed significant differences between the groups (*p* < 0.05). Based on the results, larger ROIs have a positive effect on the performance of the models where ROI BIG and ROI MED showed similar results when compared to each other (Dunn’s test, *p* = 0.1297). However, the best-performing models were achieved using the ROI BIG that is also observed in Table 5. In contrast, ROI SMALL has a clear negative impact on the average performance of the models when compared to the larger ROIs (Dunn’s test, *p* < 0.05). The results imply that estimation of workability requires a wider view of the concrete surface where the motion and interaction with the mixer can be captured. Hence, a more complex motion of the concrete surface should be captured with the depth camera.

## 4. Discussion 

The traditional methods for assessing concrete workability are labor-intensive and time-consuming. Automated measurement systems utilizing ML present an opportunity for real-time monitoring, enabling the assessment of all batched concrete. Recent investigations have focused on novel CV techniques for evaluating fresh concrete, often employing complex, computationally demanding deep-learning models with limited transparency. In contrast, our study introduces a transparent and lightweight framework that leverages depth data captured in concrete mixers to estimate slump class. Our approach not only simplifies the estimation process but also provides insight into the visual and rheological characteristics reflected in the data.

Our framework offers several advantages over both conventional methods and CV techniques employing visible light. These systems require adequate lighting and are sensitive to shadows cast on the concrete surface, which complicates model performance and demands precise light source placement. Low illumination levels can fail to freeze the motion of the mixer, resulting in blurred images. Furthermore, they capture fine textural details that vary with concrete quality, posing a risk of model overfitting. In contrast, our stereovision camera framework, which uses internal IR projection, eliminates the need for external lighting. Since the IR source is located next to the camera lenses, it minimizes the variation caused by the lighting conditions and placement of the lights and the camera. However, it must be noted that infrared technology can be less effective in environments with high ambient infrared light, such as direct sunlight. 

Based on the investigation, depth data contains essential information to predict the workability of the mixer. The trained ML classifiers predicted the broad range of slump classes, covering very stiff (S1) to very fluid concrete (S5). The ML classifiers trained with XGBoost and MPL algorithms achieved good testing accuracies of 0.8179 and 0.8078, respectively. Notably, most misclassifications occurred between adjacent slump classes, caused by the ordinal nature of the slump classes. This phenomenon can also be observed from the testing metric of QWKs, showing higher values of 0.9156 and 0.9122 for the models XGBoost and MPL, respectively. In operational settings, continuous depth data streams would mitigate the impact of misclassifications with single frames, enhancing the prediction precision, whether it involves classification or regression. 

Data collection and preparation cause risks during the development of new ML frameworks. Hence, the collected data and its preparation pipeline were first investigated using PCA. Our initial analysis with PCA affirmed that the HTFs captured essential variance within the data, with most HTFs substantially contributing to this variance. By examining the directions and magnitudes of the HTFs, we established a connection between the visual characteristics of concrete and its HTFs, observing changes in entropy and contrast corresponding to the fluidity of the concrete. It was seen that as the concrete became more fluid, the average entropy decreased and the high variation in contrast decreased. Interestingly, stiffer concretes proved more challenging for the models, likely due to their limited deformation and greater height variability without external forces. When the yield strength of the concrete surpasses a certain limit, the concrete starts to deform continuously, possibly making the prediction more predictable, and causing less variation between the frames. 

Further analysis of data preparation parameters (DM, FM, and ROI) demonstrated the robustness of the framework. Additional denoising with DM 2 did not significantly enhance model performance, which could also indicate the sufficiency of the original depth data quality. While the average accuracies with the different FMs did not improve substantially, focusing on the scoop blade with FM 2 increased the accuracy with high-performing models. Therefore, the models appear to benefit from the interaction between the concrete and the scoop blade, increasing both the accuracy and the effectiveness of the model. A larger ROI was shown to be crucial for performance, hinting that capturing expansive concrete movement is necessary for robust predictions. In other words, aiming at a smaller area may not give enough information to make reliable estimations. The angle of view was limited by the wide-angle lens, which led to cropping the frames to obtain different ROI sizes, reducing the spatial resolution of frames before conversion to HTFs. However, optimizing a suitable angle of view and spatial resolution could improve depth data quality in future pipelines.

The authors aim to deploy our framework in industrial settings to examine its applicability across various mixer types and batch properties. We plan to transition from classifying slump classes to predicting continuous slump values with regression models, which would benefit from the increased data available in such environments. An on-site investigation also allows for optimizing the framework and tuning the models as the amount of data vastly increases. We also anticipate integrating our depth camera approach with frameworks developed by other researchers to further refine and enhance the performance of our models in situ.

## 5. Conclusions 

This paper introduced a novel ML framework utilizing an active stereovision camera within a concrete mixer to estimate the slump of concrete. The collected depth data were prepared for the ML models by denoising the depth frames and extracting the core features. Additionally, the extracted frames were converted into HTFs that were used to train and evaluate the nine ML classifiers. The key findings from this study are summarized as follows:The framework demonstrated good accuracy where the XGBoost classifier achieved the highest testing accuracy of 0.8179. In addition, MLP, GBoost, and RF provided competitive results. Misclassifications typically occurred between adjacent slump classes, attributable to the ordinal nature of concrete workability. Based on the confusion matrices, the models demonstrated slightly better classification accuracy with fluid, which is likely due to their lower yield stress enabling more consistent deformation during the mixing.PCA identified the slump classes as overlapping clusters within the 2D space, formed by the two PCs. Stiff concretes were associated with more dispersed clusters, while fluid concretes generally led to more compact clusters, aligning with the findings with confusion matrices where the accuracy improved with fluid concrete. The PCA also highlighted that the HTFs significantly influenced the variance captured by the PCs, indicating a strong connection between Haralick features and dataset variance.Comparative and statistical analyses of three dataset preparation parameters (DM, FM, and ROI) revealed that the classifiers were generally robust to these variations. Specific findings include:
Denoising: The denoising of the depth data was analyzed by implementing two levels of denoising. The substantial denoising (DM 2) did not clearly impact the accuracy of the classifiers statistically (Mann–Whitney U test, *p* = 0.5750), suggesting that the imperfections in the depth data minimally impact classifier performance. However, the placement of the stereovision camera only showed a minor degradation of the collected depth data.Filtering: Four filtering methods were used to extract mixer features from the depth data. The results indicate that the filtering method (FM) can significantly impact the average model performance (Kruskal–Wallis test, *p* = 0.0046) while also requiring less data to achieve equivalent or better performance, making the framework more efficient. Interestingly, focusing on the action of the scoop blade (FM 2) increased significantly (Dunn’s test, *p* = 0.0049) the accuracy, which was most notable with the best-performing classifiers.Region of Interest: The impact of ROI selection was also investigated by comparing three crop sizes. The best performance was achieved with the larger ROIs (Kruskal–Wallis and Dunn’s test, *p* < 0.05), indicating the importance of collecting the depth data from a larger concrete surface. In addition, a larger ROI reduces the sensitivity for the placement of the camera.

## Figures and Tables

**Figure 1 sensors-24-04472-f001:**
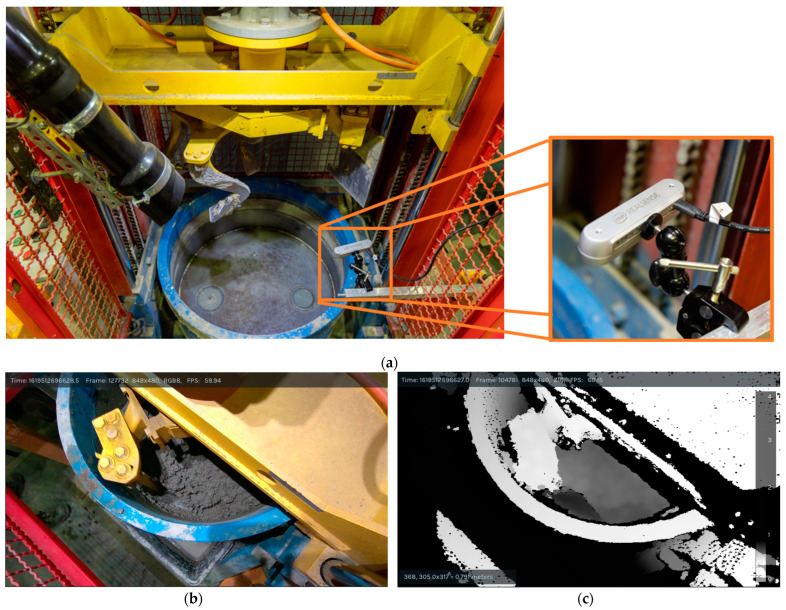
(**a**) An overview image of the planetary mixer and location of the stereovision camera. (**b**) RGB color view from the stereovision camera (**c**) Depth view from the stereovision camera with an applied grayscale color map.

**Figure 2 sensors-24-04472-f002:**
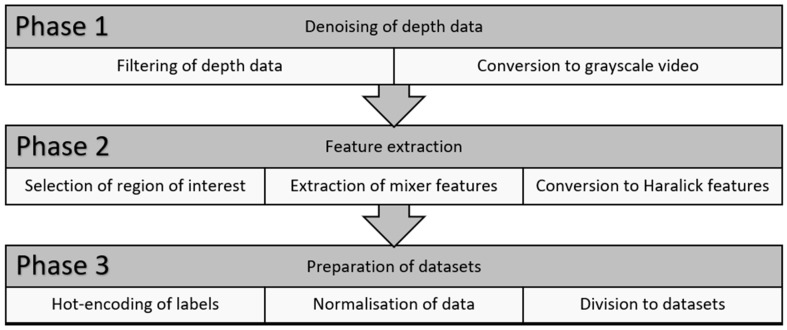
Overview of data preparation pipeline in the framework.

**Figure 3 sensors-24-04472-f003:**
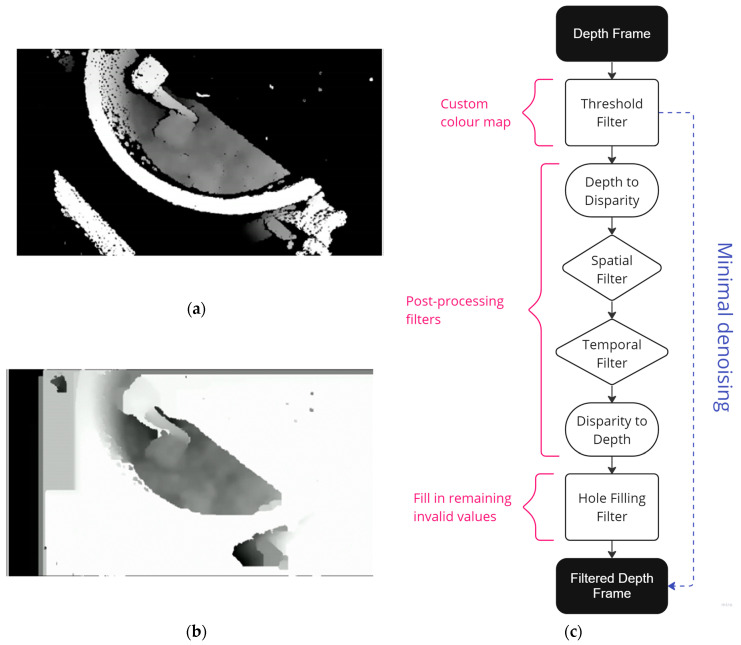
An overview of the denoising of the depth data. (**a**) An output of the minimal denoising where only a color map was applied to the data. (**b**) An output of the substantial denoising where various filters were applied to denoise the depth frame. (**c**) The flowchart shows the denoising process of the depth frame into the filtered depth frame.

**Figure 4 sensors-24-04472-f004:**
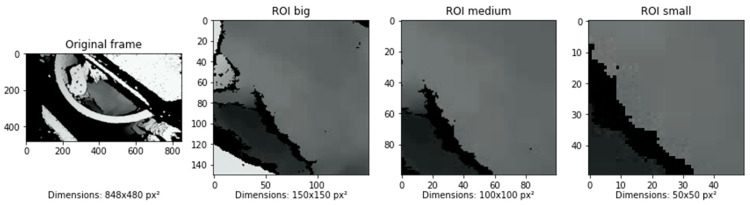
Each frame was cropped to a square based on the region of interest (ROI) size. While the largest ROI barely covered the opening of the mixer, the smallest ROI focused on the view of the trail of the mixer blades.

**Figure 5 sensors-24-04472-f005:**
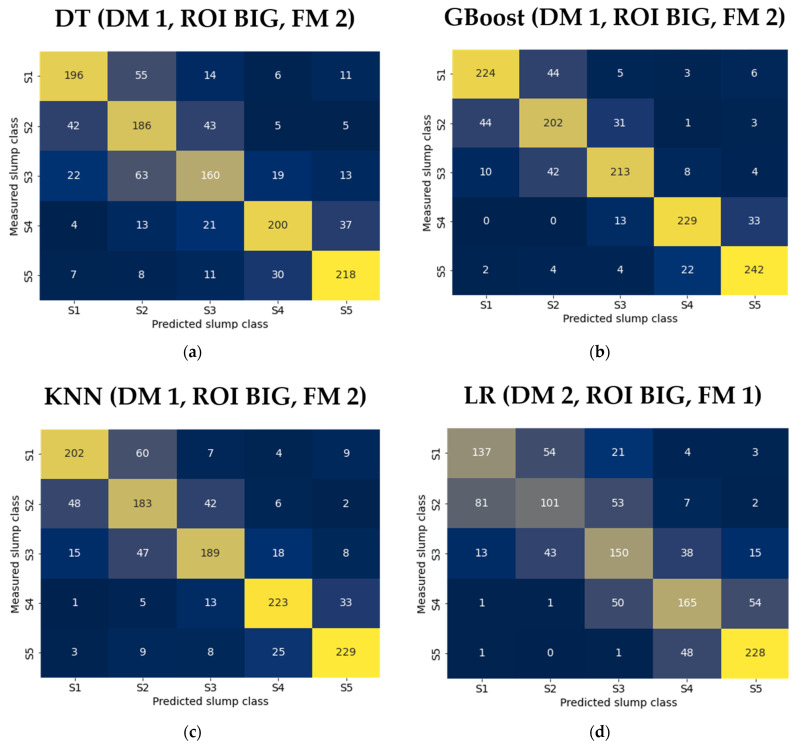
Confusion matrices of nine ML classifiers. The horizontal and vertical axes represent the predicted and actual (measured) classes, respectively. ML classifiers where: (**a**) DT = decision tree; (**b**) gradient boosting = GBoost; (**c**) KNN = k-nearest neighbor; (**d**) LR = logistic regression; (**e**) MLP = multi-layer perceptron; (**f**) NB = naïve bayes; (**g**) RF = random forest; (**h**) SVM = support vector machine; and (**i**) extreme gradient boosting (XGBoost). The dataset preparation parameters were DM: denoising method, ROI: region of interest, and FM: filtering method.

**Figure 6 sensors-24-04472-f006:**
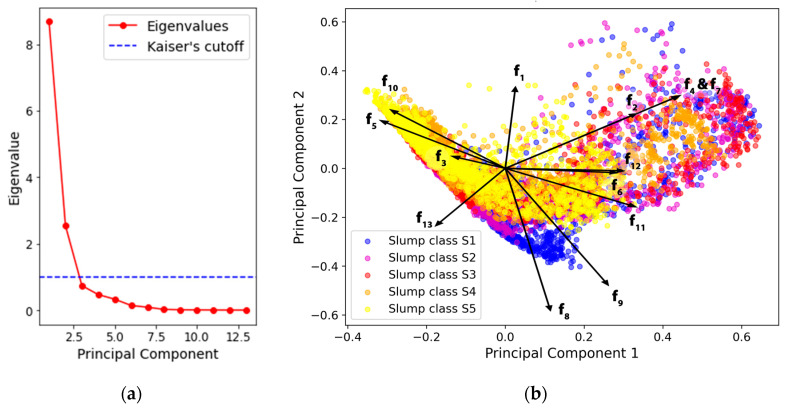
(**a**) The scree plot showing eigenvalues of the 13 principal components. (**b**) A biplot showing the influence of the computed Haralick texture features on the two principal components with the five slump classes.

**Figure 7 sensors-24-04472-f007:**
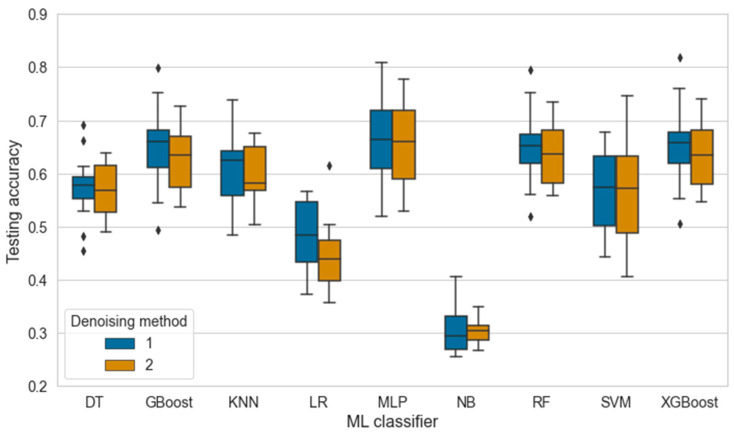
The effect of the denoising method on the accuracy of nine ML classifiers where DT = decision tree, GBoost = gradient boosting, KNN = k-nearest neighbor, LR = logistic regression, MLP = multi-layer perceptron, NB = naïve bayes, RF = random forest, SVM = support vector machine, and XGBoost = extreme gradient boosting. The outliers are marked with a diamond symbol.

**Figure 8 sensors-24-04472-f008:**
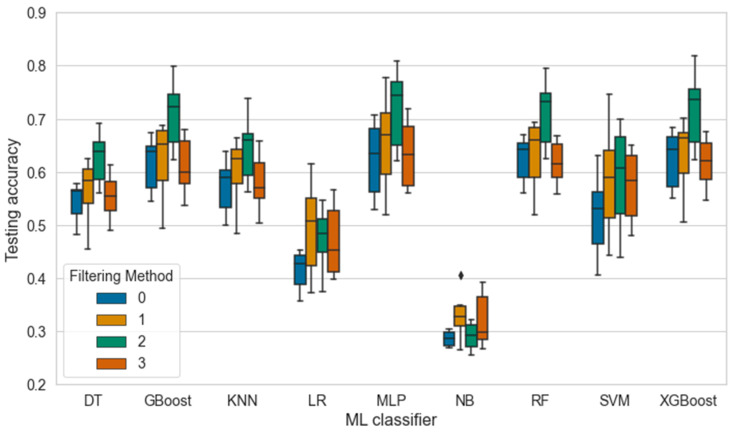
The effect of the filtering method on the accuracy of nine ML classifiers where DT = decision tree, GBoost = gradient boosting, KNN = k-nearest neighbor, LR = logistic regression, MLP = multi-layer perceptron, NB = naïve bayes, RF = random forest, SVM = support vector machine, and XGBoost = extreme gradient boosting. The outliers are marked with a diamond symbol.

**Figure 9 sensors-24-04472-f009:**
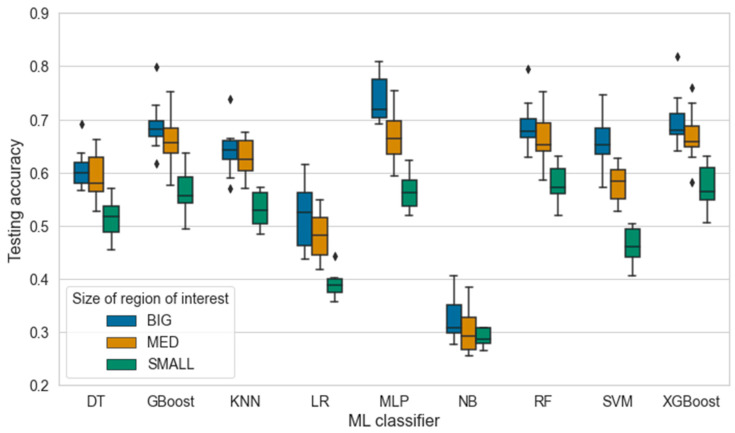
The effect of the region of interest on the accuracy of nine ML classifiers where DT = decision tree, GBoost = gradient boosting, KNN = k-nearest neighbor, LR = logistic regression, MLP = multi-layer perceptron, NB = naïve bayes, RF = random forest, SVM = support vector machine, and XGBoost = extreme gradient boosting. The outliers are marked with a diamond symbol.

**Table 1 sensors-24-04472-t001:** The description and definition of the gray-level co-occurrence matrix (GLCM) and its variables 1–4 and the descriptions and definitions to compute the 13 Haralick texture features with its required variables 5–6 based on [40,41].

**Description**	Definition	
Normalized GLCM	$p(i,j)=\frac{P\left(i,j\right)}{\sum_{i=1}^{N_{g}}\sum_{j=1}^{N_{g}}P(i,j)}$	(2)
Variable 1	$p_{x}(i)=\sum_{j=1}^{N_{g}}p(i,j)$	(3)
Variable 2	$p_{y}(j)=\sum_{i=1}^{N_{g}}p(i,j)$	(4)
Variable 3	$p_{x+y}(k)=\sum_{i=1}^{N_{g}}\sum_{j=1}^{N_{g}}P\left(i,j\right),k=2,3,\ldots ,2N_{g}$	(5)
Variable 4	$p_{x-y}\left(k\right)=\sum_{i=1}^{N_{g}}\sum_{j=1}^{N_{g}}P\left(i,j\right),k=0,1\ldots N_{g}-1$	(6)
(1) Angular Second Moment	$f_{1}=\sum_{i=1}^{N_{g}}\sum_{j=1}^{N_{g}}{\left\{p\left(i,j\right)\right\}}^{2}$	(7)
(2) Contrast	$f_{2}=\sum_{k=0}^{N_{g}-1}k^{2}p_{x-y}k$	(8)
(3) Correlation	$f_{3}=\frac{\sum_{i=1}^{N_{g}}\sum_{j=1}^{N_{g}}\left(ij\right)p\left(i,j\right)-\mu_{x}\mu_{y}}{\sigma_{x}\sigma_{y}}$	(9)
where	$\mu_{x},\;\mu_{y},\;\sigma_{x},$ and $\sigma_{y}$ represent the means and standard deviation of $p_{x}$ and $p_{y}$.	
(4) Sum of Squares: Variance	$f_{4}=\sum_{i=1}^{N_{g}}\sum_{j=1}^{N_{g}}{\left(i-\mu \right)}^{2}p(i,j)$	(10)
(5) Inverse Difference Moment	$f_{5}=\sum_{i=1}^{N_{g}}\sum_{j=1}^{N_{g}}\frac{1}{1+{\left(i-j\right)}^{2}}p(i,j)$	(11)
(6) Sum Average	$f_{6}=\sum_{i=2}^{{2N}_{g}}ip_{x+y}(i)$	(12)
(7) Sum Variance	$f_{7}=\sum_{i=2}^{{2N}_{g}}{\left(i-f_{8}\right)}^{2}p_{x+y(i)}$	(13)
(8) Sum entropy	$f_{8}=-\sum_{i=2}^{{2N}_{g}}p_{x+y}(i)\log\left(p_{x+y}(i)\right)$	(14)
(9) Entropy	$f_{9}=-\sum_{i=1}^{N_{g}}\sum_{j=1}^{N_{g}}p(i,j)\log\left(p(i,j)\right)$	(15)
(10) Difference Variance	$f_{10}=variance\;of\;p_{x-y}$	(16)
(11) Difference Entropy	$f_{11}=-\sum_{i=0}^{N_{g}-1}p_{x-y}(i)\log\left(p_{x-y}(i)\right)$	(17)
(12) Information Measure of Correlations	$f_{12}=\frac{f_{9}-HXY1}{\max\left\{HX,HY\right\}}$	(18)
(13) Information Measure of Correlations	$f_{13}={\left(1-e^{\left(-2.0(HXY2-HXY\right)}\right)}^{1/2}$	(19)
where	HX and HY are entropies of $p_{x}$ and $p_{y}$.	
Variable 5	$HXY1=-\sum_{i=1}^{N_{g}}\sum_{j=1}^{N_{g}}p(i,j)\log\left(p_{x}\left(i\right)p_{y}(j)\right)$	(20)
Variable 6	$HXY2=-\sum_{i=1}^{N_{g}}\sum_{j=1}^{N_{g}}p_{x}\left(i\right)p_{y}(j)\log\left(p_{x}\left(i\right)p_{y}\left(j\right)\right)$	(21)

**Table 2 sensors-24-04472-t002:** Parameters of the dataset preparation and the number of observation points in each dataset.

Denoising Method	Region of Interest	Filtering Method	Total Number of Observation Points	Number of Training Observation Points	Number of Testing Observation Points
DM 1	ROI BIG	FM 0	12,000	9600	2400
		FM 1	2854	2283	571
		FM 2	6939	5551	1388
		FM 3	6416	5132	1284
	ROI MED	FM 0	12,000	9600	2400
		FM 1	4412	3529	883
		FM 2	6939	5551	1388
		FM 3	6416	5132	1284
	ROI SMALL	FM 0	12,000	9600	2400
		FM 1	5052	4041	1011
		FM 2	6939	5551	1388
		FM 3	6416	5132	1284
DM 2	ROI BIG	FM 0	12,000	9600	2400
		FM 1	6339	5071	1268
		FM 2	6978	5582	1396
		FM 3	6382	5105	1277
	ROI MED	FM 0	12,000	9600	2400
		FM 1	7382	5905	1477
		FM 2	6978	5582	1396
		FM 3	6382	5105	1277
	ROI SMALL	FM 0	12,000	9600	2400
		FM 1	8117	6493	1624
		FM 2	6978	5582	1396
		FM 3	6382	5105	1277

**Table 3 sensors-24-04472-t003:** Selected machine learning classifiers and their working principles.

ML Classifier	Abbreviation	Working Principle
Decision Tree	DT	It is a set of hierarchical tests where the final decision or outcome is drawn from the terminal node [45].
Gradient Boosting	GBoost	Combines multiple weak models (usually decision trees) to create a strong predictive model that does not use regularization [46].
K-Nearest Neighbor	KNN	Predicts the class of an unseen point by voting by finding the k-nearest neighbors and assigning the point the same label as the most voted label [47].
Logistic Regression	LR	Finds the linear decision boundaries that separate the differing classes [48,49]. A ℓ2-regularization [50] technique was applied automatically.
Multi-layer Perceptron	MLP	A feedforwarding artificial neural network that is comprised of fully connected neurons [51].
Naïve Bayes	NB	Information in the dataset is used to estimate the posterior probability of each class y given object x, which is then used for classification purposes [52,53].
Random Forest	RF	Instead of a single decision tree, an ensemble of multiple trees is trained on the dataset [45,54],
Support Vector Machine	SVM	establishing optimal hyperplanes to differentiate classes in data using linear algebra [55].
eXtreme Gradient Boosting	XGBoost	Develops a series of weak learners by aggregating the predictions of several weak models, such as decision trees [56] that regarded as optimized and scalable version of GBoost with regularization [17].

**Table 4 sensors-24-04472-t004:** The results of slump measurements and their corresponding slump classes. The allowed slump within slump class represents the range of slump values used to categorize the slump into slump classes S1 to S5.

Cycle No.	Slump [mm]	Slump Class [-]	Allowed Slump within the Slump Class [mm]
1	30	S1	10–40
2	70	S2	50–90
3	150	S3	100–150
4	170	S4	160–210
5	200	S4	160–210
6	230	S5	≥220

**Table 5 sensors-24-04472-t005:** Evaluation results based on the testing dataset. The following columns are represented: Classifier denotes the used ML classifier algorithm, DM denotes the denoising method, ROI denotes the region of interest, FM denotes the filtering method and QWK denotes quadratic weight kappa.

Classifier	DM	ROI	FM	Testing Accuracy	Testing Precision	Testing Recall	Testing F1 Score	Testing QWK
DT	1	BIG	2	0.6911	0.6945	0.6911	0.6916	0.8000
GBoost	1	BIG	2	0.7991	0.8002	0.7991	0.7994	0.9017
KNN	1	BIG	2	0.7387	0.7405	0.7387	0.7391	0.8529
LR	2	BIG	1	0.6145	0.6093	0.6145	0.6104	0.8458
MLP	2	BIG	1	0.8078	0.8110	0.8078	0.8081	0.9122
NB	1	BIG	1	0.4049	0.4132	0.4049	0.4019	0.6246
RF	1	BIG	2	0.7855	0.7951	0.7955	0.7951	0.8787
SVM	2	BIG	1	0.7467	0.7455	0.7467	0.7456	0.8963
XGBoost	1	BIG	2	0.8179	0.8184	0.8179	0.8179	0.9156

## Data Availability

The data presented in this study are available on request from the corresponding author.

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
