# Peer review of "Estimating the Workability of Concrete with a Stereovision Camera during Mixing"

_sensors, 2024, doi:10.3390/s24144472_

Round 1
Reviewer 1 Report
Comments and Suggestions for Authors
This manuscript aims to automate the workability of concrete by means of a Computer Vision technique.
The contribution of this study is not significant enough to be published in a journal yet.
My comments to improve the content of this manuscript are as follows:
1. The proposed method should be novel in contribution to science and technology. Please provide a detailed discussion about contrasting with the state of the art.
2. Besides the other performance measures specificity and AUC should also be reported for model evaluation.
3. The authors proposed some ad hoc engineered texture features. It is well known that the texture features are susceptible to the variations in the lighting environment. Authors should discuss robustness of the system under the uncertainty and invariance characteristics of the features utilized.
4. The PCA analysis and performance results suggest that either the ML model architecture/hyperparameters are not well-tuned, or the feature-based representation, or both, do not distinctly describe the nature under investigation. Authors should improve the system by conducting hyperparameter tuning as well as proposing distinctive features.
Reviewer 2 Report
Comments and Suggestions for Authors
The Haralick features and the seven selected classifiers are well-known within the field. The authors tested the methods separately and provide the results. However, the results are unsatisfactory, and they did not propose new methods to boost the performance.
